# Impact on Glycemic Variation Caused by a Change in the Dietary Intake Sequence

**DOI:** 10.3390/foods12051055

**Published:** 2023-03-02

**Authors:** Alexis Alonso-Bastida, Manuel Adam-Medina, Dolores-Azucena Salazar-Piña, Ricardo-Fabricio Escobar-Jiménez, María-Socorro Parra-Cabrera, Marisol Cervantes-Bobadilla

**Affiliations:** 1TecNM/CENIDET, Electronic Engineering Department, Interior Internado Palmira S/N, Palmira, Cuernavaca 62490, Mexico; 2Faculty of Nutrition, UAEM, Cuernavaca 62350, Mexico; 3Population Health Research Center, National Institute of Public Health, Cuernavaca 62100, Mexico; 4Center of Research in Engineering and Applied Sciences (CIICAp-IICBA)/UAEM, Av. Universidad 1001, Chamilpa, Cuernavaca 62209, Mexico

**Keywords:** diet assistance, glucose curve, glucose homeostasis, glycemic variability, healthy person, macronutrient intake sequence

## Abstract

This work presents an analysis of the effect on glycemic variation caused by modifying the macronutrient intake sequence in a person without a diagnosis of diabetes. In this work, three types of nutritional studies were developed: (1) glucose variation under conditions of daily intake (food mixture); (2) glucose variation under conditions of daily intake modifying the macronutrient intake sequence; (3) glucose variation after a modification in the diet and macronutrient intake sequence. The focus of this research is to obtain preliminary results on the effectiveness of a nutritional intervention based on the modification of the sequence of macronutrient intake in a healthy person during 14-day periods. The results obtained corroborate the positive effect on the glucose of consuming vegetables, fiber, or proteins before carbohydrates, decreasing the peaks in the postprandial glucose curves (vegetables: 113–117 mg/dL; proteins: 107–112 mg/dL; carbohydrates: 115–125 mg/dL) and reducing the average levels of blood glucose concentrations (vegetables: 87–95 mg/dL; proteins: 82–99 mg/dL; carbohydrates: 90–98 mg/dL). The present work demonstrates the preliminary potential of the sequence in the macronutrient intake for the generation of alternatives of prevention and solution of chronic degenerative diseases, improving the management of glucose in the organism and permeating in the reduction of weight and the state of health of the individuals.

## 1. Introduction

Consumption of food in human beings is an activity that provides the nutrients necessary for the adequate performance of the organism and prevents various diseases such as diabetes and cardiovascular diseases [1]. Consumption of any food generates elevations in glucose levels, defined as “glucose curves,” because of a gradual elevation in blood glucose levels that is subsequently attenuated due to the homeostatic processes of glucose in the organism throughout time [2]. This period is called the postprandial glucose stage, which lasts 4–5 h for each meal taken [3]. Information on the magnitude, fluctuations, and different characteristics of glucose curves (peaks, plateaus, rise and decay times) is defined as “glycemic variability” [4], which has taken on great relevance for the generation of actions toward the development of nutritional interventions.

Prolonged postprandial glucose episodes and their high frequency generate one of the main risk factors for developing Type 2 Diabetes Mellitus (T2DM) [5,6] since the average blood sugar level throughout the day is above the basal glucose levels. T2DM is a multifactorial disease occurring in the adult population [7], where the main characteristic is the presence of elevated blood glucose levels throughout the day (episodes of hyperglycemia) [8]. High glucose levels are mainly related to insulin resistance, a condition in which the different cells of the organism cannot assimilate the insulin hormone adequately. This condition leads to an increase in insulin secretion by the pancreas, which in turn facilitates the presence of hyperglycemia in the organism due to the low absorption of insulin by the cells [9]. This set of diseases is usually the product of carrying out many habits detrimental to health throughout the person’s life before being diagnosed with T2DM [10].

T2DM is a chronic degenerative disease that leads to a degradation in the quality of life and facilitates the presence of cardiovascular complications [11] and renal, ocular, and liver diseases [12], these being a small part of the set of conditions related to T2DM. It should be noted that because of the disease control actions by the health sector, there is an added problem [13,14], ranging from the viewpoint of proper care for the community [15] to the economic aspect where the investment required to satisfy this need is growing every year [16,17]. Therefore, several alternatives have been developed to prevent the condition, some of them focused on informing society about how a healthy diet and regular physical activity reduce the chances of developing T2DM [18].

Diet is the main factor in the increase in glucose levels, so it is essential to control how we consume food. Several studies have addressed this problem, thus generating alternative solutions to reduce the impact of postprandial glucose, such as Dahl et al. [19], who demonstrated how semaglutide has beneficial effects on the reduction of postprandial glucose, triglycerides, glucagon, and gastric emptying in people with T2DM. Rayner et al. [20] similarly demonstrate the effects of lixisenatide in reducing gastric emptying by promoting postprandial glucose dynamics. Vlachos et al. [21] present an in-depth review of the subject, concluding that reducing carbohydrates in conjunction with a higher fiber intake positively affects postprandial glucose reduction.

How different macronutrients are ingested affects the glycemic variability of organisms, modifying the time to glucose elevation, the glucose curve magnitude, and the glucose decay time [22,23]. Sun et al. [24] developed a study of 16 healthy people in which the effect of the proper order for macronutrient intake on glycemic variation was evaluated. The study showed that consuming vegetables followed by proteins and concluding with carbohydrates is an effective strategy to reduce postprandial glucose and prevent the generation of T2DM. Kubota et al. [25] corroborate that the correct order of the food sequence reduces the episodes of postprandial hyperglycemia, improving weight loss and metabolic function.

Considering the alternatives presented in the various sources of information and considering that those related works generate an analysis on the modification of the sequence of macronutrient intake concerning tests of about 4 to 5 h of glucose monitoring, the main objective of this work is to obtain preliminary results on the effectiveness of a nutritional intervention based on the modification of the order of macronutrient intake in a healthy person. The particularity of this work is to generate continuous glucose measurements during three periods of 14 days in an individual for whom three types of nutritional interventions were developed. In this way, difficult-to-access information is obtained on the behavior of glucose curves derived from the proposed nutritional intervention. Considering the scope of the present work, this research is a first approximation for the development on a larger scale of nutritional interventions focused on the modification of the macronutrient sequence that allows the reduction of postprandial glucose levels in healthy people. This way, the necessary conditions are obtained to carry out this experimentation on a larger scale. The results of this research will allow the generation of information for the development of alternative solutions in the generation of metabolic and chronic degenerative diseases.

## 2. Materials and Methods

### 2.1. Quasiexperimental Study Design

The study methodology consists of the steps described below:Generation of data: A series of body measurements, an indirect calorimetry test, and the development of a food reminder were developed in the participant to have an approximation of the nutritional status of the participant and to be able to propose the type of interventions in the sequence of macronutrient intake to follow so that there is no decompensation in the current type of food intake.Implantation of the continuous glucose monitoring sensor: In each test, a new interstitial glucose sensor was implanted to generate data on glucose dynamics.Daily diet (Test 1): Subsequently, Test 1 was developed, where glucose measurements were generated and focused on describing the variation of glucose levels in the face of the study subject’s daily diet (mixture of macronutrients without having any order in food consumption).Regular diet with ordered consumption of macronutrients (Test 2): Test 2 has the objective of obtaining the glucose dynamics when the sequence in the order of macronutrient consumption is modified without generating any change in the participant’s regular diet.Assisted diet with ordered consumption of macronutrients (Test 3): This test consists of generating measurements of glucose variation in the face of a modification in the participant’s daily diet considering the change in the sequence of macronutrient intake.Statistical analysis: Once the three different tests were generated, a statistical analysis of the results obtained was generated, which was the study’s core. In this analysis, the impact of the sequence in macronutrient consumption was quantified and contrasted concerning the postprandial glucose curves generated in each dietary intake. For this purpose, the proportions of macronutrients consumed per intake were used and related to glucose concentrations, magnitudes of the postprandial glucose peaks, and times in which postprandial glucose stabilizes.

### 2.2. Ethics of Research

In the research developed, the health and integrity of the participant were not put at risk in any way, being an observational experiment. Each of the procedures developed was evaluated and authorized by the Ethics Committee of the Faculty of Medicine of the Autonomous University of the State of Morelos (CONBIOETICA-17-CEI-003-201-81112). It should be emphasized that the participant was informed of the procedures to be developed, and once informed and in agreement with the guidelines to be developed, the participant signed the letter of informed consent.

### 2.3. Instrumentation

The instrumentation used in this study consists of an interstitial continuous glucose monitoring (CGCM) system (Freestyle libre, Abbott^®^, Chicago, IL, USA) for the acquisition of glucose measurements, an indirect calorimetry system (KORR Medical^®^, West Valley City, UT, USA) to obtain the participant’s daily energy consumption, a bioimpedance scale (BC-545 Segmental, Tanita^®^ brand, Arlington Heights, IL , USA) and a stadiometer for the participant’s body detection, and a food intake and physical activity diary for macronutrient counting and physical activity intensity.

There are no conflicts of interest in this research. There is no relationship between the suppliers of the instrumentation used in the experimentation. The instruments were purchased with funding from CONACYT (project number 320155) and TecNM (project numbers 14002.22P and 14003.22P).

### 2.4. Subject of Study

The proposed study has great difficulty in its development in population studies due to the strict discipline required to carry out the dietary sequence in the required order, the filling of the food intake and physical activity diaries, and the glucose monitoring. Therefore, this study was developed on a pilot basis in a physically active healthy person (without a diagnosis of diabetes or any chronic degenerative disease) considered by the standards of a healthy person proposed by the World Health Organization (WHO) [26]. The participant is a 26-year-old male with a height of 1.78 m and a daily energy intake of 2810 calories (246 calories from physical activity, 591 calories derived from the participant’s daily activities and lifestyle, and 1973 calories from energy consumed at rest). Body measurements were taken at the beginning of each test, described in Table 1, where weight, body mass index, abdominal circumference, and percentages of muscle, fat, and visceral fat are considered.

To have certainty in the information, the participant received instructions to correctly fill out the food intake and physical activity recording instruments (both devices are standardized forms). In addition, the participant was instructed to record foods that were not consumed or added to the tools. This was combined with a 24-h reminder of the food consumed, carried out by trained personnel. Regarding continuous glucose monitoring, due to the conditions of the measuring instrument, the patient was instructed to take periodic manual measurements throughout the day, avoiding more than four hours between sizes (except for the participant’s sleeping hours). The correct storage of glucose readings was corroborated with the report generated by the Abbott^®^ platform. For more information on the monitoring system, we recommend consulting [27].

### 2.5. Food Sequence

Considering the three tests developed, in the case of Tests 2 and 3, the participant was assigned a sequence in the intake of macronutrients, sectioned according to the type of intake developed (breakfast, snack 1, lunch, dinner, and snack 2) and repeating the type of food and its quantity for 4 consecutive days exchanging the order in the consumption of macronutrients. (This is because the monotony of the food makes it difficult for participants to adhere to the needs of the experiment.) The sequence is presented regarding the symbology of the proposed macronutrients starting with the symbol on the left side and ending with the symbol on the right side (VF-CH-P-FT = 1. Vegetables and Fiber, 2. Carbohydrates, 3. Proteins of animal origin, 4. Fats). Table 2 presents the sequence used for each day of intake, denoting macronutrients as follows: P: Proteins of animal origin; CH: Carbohydrates; VF: Vegetables and Fiber; D: D: Dairy; FT: Fats; FR: Fruits.

### 2.6. Proportions of Macronutrients Ingested

Each food ingested by the participant was analyzed concerning its composition in carbohydrates, lipids, and proteins, obtaining for each macronutrient the weight (grams) and energetic quantity (calories) contained in the food. In addition, the percentage of energy provided by each macronutrient in each of the intakes analyzed was calculated. Table 3 presents the mean and standard deviations of the composition of each macronutrient analyzed in each intake developed for each of the three types of diets analyzed.

### 2.7. Diet for Each Test Developed

Throughout the experimentation, several menus were used to ensure proper adherence to the study by the participant. In the case of Tests 2 and 3, each of the menus was appropriately developed in such a way that the proportions of macronutrients ingested were analogous. For the reader to have a clear idea of the menu composition, the following is an example for each of the menus developed in each test.

Test 1: Breakfast, smoothie with the following items, 1 banana (100 g), 33 g amaranth, 84 g oatmeal, and milk (500 mL), 2 eggs, and 2 corn tortillas (developed at 7:50 h). Lunch, 15 g of chickpeas, a piece of bread, and 500 mL of water (developed at 13:10 h). Snack 1, smoothie with the following items, 25 g whey protein, and 350 mL of milk (developed at 19:00 h). Dinner, 1 sandwich consisting of 40 g turkey ham and 25 g of cheese, and 350 mL of milk (developed at 19:15 h). Snack 2, 30 peanuts (developed at 21:00 h).Test 2: Breakfast, 240 mL of whole milk, 2 scrambled eggs, 2 corn tortillas, and 500 mL of water (developed at 7:40 h). Snack 1, a smoothie with the following items, 25 g whey protein, 84 g oatmeal, 1 banana (100 g), and 500 mL of water (developed 13:20 h). Snack 2, 3 nuts (developed at 21:30 h). Lunch, a salad (lettuce and tomato), 60 g rice, 100 g beef steak, 3 corn tortillas, 15 peanuts, and 500 mL of water (developed at 15:30 h). Dinner, 60 g chicken breast, 120 g salad (potato, carrot, and pea), 50 g jicama, 18 pieces of grapes, and 500 mL water (developed at 20:30 h).Test 3: Breakfast, 2 slices of white bread, 40 g of turkey ham, 60 g of salad (alfalfa sprouts, lettuce, and tomato), 23 g avocado, and 500 mL of water (developed at 8:15 h). Snack 1, 25 g whey protein, 100 g Greek yogurt, and 110 g apple (developed at 11:30 h). Lunch, 60 g chicken, 60 g bell peppers and onion, 2 tortillas, 100 g, and 1000 mL water (developed at 16:15 h). Dinner, 2 baked corn tostadas, 90 g tuna, 50 g onion and tomato, 23 g avocado, and 750 mL of water (developed at 19:00 h). Snack 2, 150 g of pineapple, 100 g of jicama, 15 peanuts, and 500 mL of water (developed at 21:40 h.).

### 2.8. Glucose Curve

The analysis of glycemic variability considers information on the dynamics of the glucose curves produced at each dietary intake. Figure 1 shows a contrast between the glucose measurement (left graph) and the respective magnitude of the curve generated after a meal (right graph). The beginning of the curve is the moment when food intake is generated, followed by the absorption of macronutrients by the organism, followed by a pronounced elevation in glucose levels until reaching the maximum peak, from where a decrease in glucose begins because of the homeostatic regulation process generated by the organism, thus generating abrupt changes in glucose derived from the effect of insulin secretion. Once the decline is complete, glucose stabilizes, thus attenuating the postprandial glucose curve generated. The magnitude of the glucose elevation (right graph) is calculated by subtracting the initial measurement from the glucose curve analyzed in each of the measurements over the time of the curve, thus having a magnitude of 0 mg/dL at the beginning, which over time can have positive or negative glucose concentration values due to the different types of absorption of the macronutrients ingested.

## 3. Results

### 3.1. Glucose Measurement

In each test, 14 days of glucose measurements were generated. The result of the glucose variation in each test is presented in Figure 2, positioning in the upper part the glucose dynamics according to a daily diet (Test 1), in the central part the dynamics according to a daily diet modifying the sequence in the macronutrient intake (Test 2), and finally, in the lower part the glucose variation according to a modification in the diet and sequence of macronutrient intake (Test 3). The difference between tests is clear according to each of the glucose curves, being greater in Test 1 since there is no fixed schedule for food intake, contrary to what happened in Test 3, where the time in the glucose curves is constant. Consequently, the behavior of glucose is more homogeneous.

Quantitatively, the average variation between each of the tests is described in Table 4, where the glucose average data, the glucose management indicator (based on that proposed by Leelarathna et al. [28]), and the glucose coefficient of variation (considering that submitted by Rodbard [29]) are presented. These data were calculated for total glucose measurements in each test performed over the 14 days of size. The results demonstrate how a higher glucose variation coefficient correlates with lower glucose concentrations and a glucose management indicator. This phenomenon is visible when comparing Test 1 results with those in Test 3.

Five types of intakes were generated for each day throughout the tests. The postprandial glucose average of each intake developed throughout every test is presented in Table 5, where the postprandial glucose concentrations are lower at breakfast (ranging between 83–89 mg/dL) due to starting from a condition close to the basal level, contrary to dinner where the glucose ranges between 95–100 mg/dL because the glucose curve starts from a higher level since the time gaps between each intake avoid the homeostasis to reach a basal level after a postprandial period.

### 3.2. Food Sequence Modification Effect on Glucose

There are marked differences between the intakes analyzed in each test. Therefore, this work explored the magnitudes, elevation, and stabilization times in the different glucose curves developed in each intake. For evaluating the impact of the sequence of macronutrients in the dietary intake, each of the glucose curves was grouped among three different types of patterns:Carbohydrate intake at the beginning;Vegetable and fiber intake at the beginning;Animal protein intake at the beginning.

The results of this analysis are presented in Table 6, highlighting the following aspects: (a) Higher maximum glucose peaks occur when carbohydrates are consumed first. (b) The consumption of vegetables and fiber or proteins generates lower glucose average levels in contrast to an early consumption of carbohydrates. (c) Early consumption of carbohydrates generates shorter periods of elevation and stabilization in the glucose curves in contrast to an early consumption of proteins or vegetables, resulting in higher glucose average levels.

As a complement to Table 6, Figure 3 illustrates the dynamics of the glucose curves when the first macronutrient ingested is carbohydrates. It shows three graphs corresponding to the three main intakes (breakfast, lunch, and dinner). Each chart has two types of colors, red (referring to the results obtained in Test 2) and blue (Test 3), representative of the dynamics of the intake developed throughout the experiment, thus illustrating the postprandial glucose during a period of 5 h of measurement. Considering the numerical results presented, the postprandial glucose dynamics show lower elevations at breakfast (maximum glucose peaks below 130 mg/dL) than at lunch, where glucose peaks reach values close to 150 mg/dL, and dinner with top mounts above 150 mg/dL.

## 4. Discussion

Recently, the concept regarding the sequence and frequency of macronutrient intake has gained strength due to the benefits it generates for the organism [30]. Paoli et al. [31] proposed an example of this, where the modification of the frequency of intakes generates benefits in the reduction of intestinal inflammation, improving autophagy, and stress resistance. Henry et al. [32] allude to how consuming vegetables before carbohydrates is a strategy capable of optimizing glycemic control and positively influencing postprandial glucose. On the other hand, King et al. [33] describe how consuming a small dose of whey protein before a macronutrient meal mix stimulates insulin generation and improves postprandial glucose in people with T2DM.

Considering the current need for the generation of information and alternative solutions for the prevention of T2DM, this work developed the necessary experimentation to determine the effects of the sequence of macronutrient intake on glucose reduction after the adoption of dietary regimens that promote the decrease in glucose levels evaluated during periods of 14 days. The results presented in a preliminary way the effectiveness of the adoption of nutritional regimens focused on the anticipated consumption of vegetables, fiber, or protein for the reduction and good condition of glucose levels. The best results came from an early intake of vegetables and fiber with an average glucose of 87–95 mg/dL and peak and stabilization times starting at 2.34 and 2.96 h, respectively. In this way, pronounced postprandial glucose episodes are avoided, and the shape of the glucose curves is flattened, in contrast to early carbohydrate intake, which has peak values of 130–150 mg/dL.

The development of the experimentation presents significant difficulty for the participant and the researcher who carries it out, because a dietary plan must be designed for each participant, thus promoting adherence to the menu and guaranteeing the correct performance of the experiment. The gradual variety of the menus is of utmost importance since it favors the participant’s comfort and decreases the probability of desertion during the experimentation. The participant’s correct development of the experiment must be corroborated with the filling of food diaries and continuous glucose measurements. In the case of sample scaling, it is advisable to consider those mentioned above, thus favoring the conditions for correct development in the experimentation. For population studies, two experimental periods with a duration of 14 days per period should be developed. In the first one, the glycemic variation is evaluated under an assisted diet, and in the second one, under a similar diet, but in this case varying the sequence in which the different types of macronutrients are ingested. It is recommended that there be a 14-day rest period between each of the tests. Otherwise, adherence to experimentation is challenging in the second experiment stage.

The complexity of this type of research illustrates the difficulties that exist for participants in adhering to a dietary regimen [34,35]. Consequently, the primary concentration of this type of research is limited to evaluating a single glucose curve in a population of healthy individuals, as is the case of Sun et al. [24] with a population of 16 healthy individuals. Alternatively, research is limited to evaluating people with gestational diabetes mellitus (GDM), as proposed in Yong et al. [36], where, like what is presented in this work, glycemic variation is analyzed in 10 women with GDM, exchanging the sequence of macronutrients and measuring glucose with a GCM. The results of this work agree with Sun et al. [24] and Yong et al. [36], where an early consumption of vegetables, fiber, or protein reduces postprandial glucose. Taking these works as a reference point, both use only one feeding plan due to a shorter duration of the experimentation, contrary to the case presented in this work, where the time of investigation makes it necessary to change the feeding plan periodically. Classifying the macronutrients consumed in these studies is similar to the method used in this research, where food is classified according to the predominant macronutrient in its composition.

The particularity of this work is focused on four specific points:Development of an analytical study on the effect of food sequence on postprandial glucose curves and the impact on glucose levels throughout the research period, with statistical analysis being a fundamental part of generating the results obtained;Experimentation time of 42 days divided into three different tests;Periodic change in meal plans to achieve patient adherence to experimentation;Contrast between three different conditions of glycemic variation derived from the tests proposed (the basis for experimentation on a more significant number of population).

The experimentation developed is a pilot test that serves as a reference to evaluate the feasibility of carrying it out in a larger population under specific conditions of degradation in glucose homeostasis. Although the measures that must be taken to develop the experimentation are extensive, the benefits gained from it are significant. These can be included in the wide range of nutritional alternatives that can be proposed for the management and prevention of diabetes. One of the main benefits is the possibility of attending to the problem without generating an extra economic cost derived from its treatment. The present work opens a window of opportunities for developing several topics focused on managing and preventing metabolic diseases from a nutritional point of view.

## 5. Conclusions

In this work, three types of nutritional studies were developed to analyze the effect of managing the order of macronutrient intake. The results are consistent with the literature, indicating that early consumption of vegetables, fiber, or protein reduces the size of the postprandial glucose curves, thus decreasing blood glucose levels and improving glucose homeostasis in the organism. Considering that the work is a pilot test, based on the results obtained and the recommendations proposed to carry out the experimentation, it is feasible to develop it on a larger sample scale. This research is a potential milestone for the generation of knowledge focused on improving glucose homeostasis in different treatments for diabetes. This work generates the possibility of creating alternatives for the prevention and control of type 2 diabetes based on changes in the dietary sequence and in conjunction with pharmacological treatment (in the case of diabetes) that does not generate an extra economic expense for the health sectors and the people treated in them.

## Figures and Tables

**Figure 1 foods-12-01055-f001:**
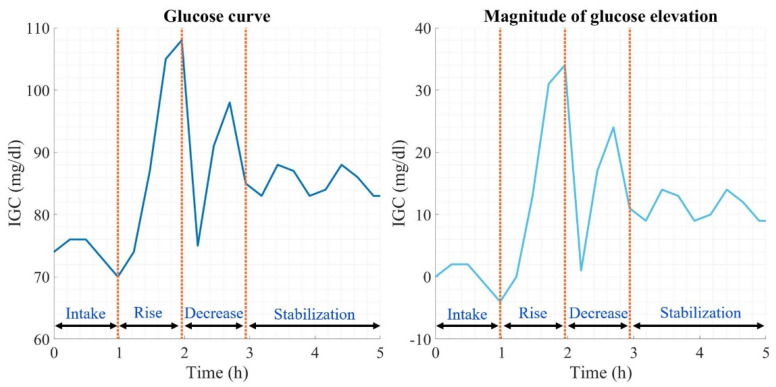
Glucose curve description.

**Figure 2 foods-12-01055-f002:**
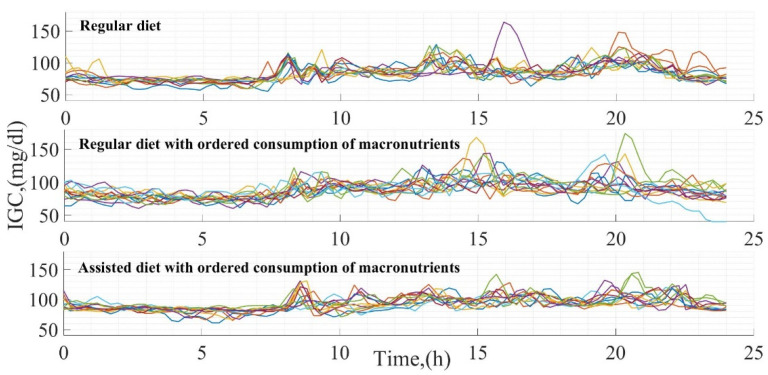
Glucose measurements in each test developed.

**Figure 3 foods-12-01055-f003:**
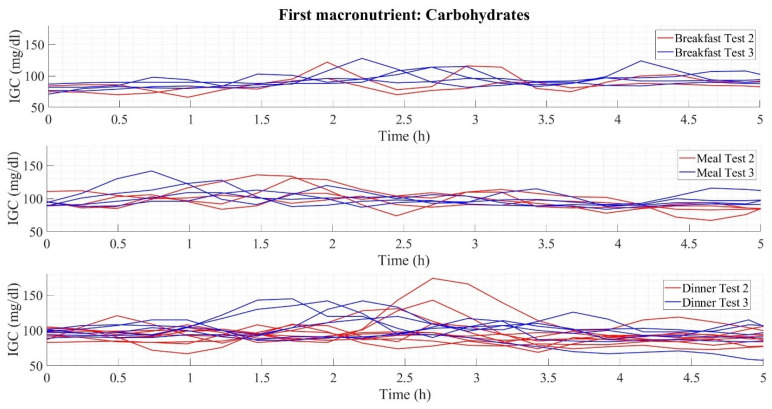
Glycemic variability in early carbohydrate consumption.

**Table 1 foods-12-01055-t001:** Subject’s body measurements at the start of each study.

Physiological Characteristics	Test 1	Test 2	Test 3
Body weight (kg)	65.4	67.1	64.6
Body mass index (kg/m^2^)	20.6	21.2	20.4
Abdominal circumference (m)	0.79	0.79	0.79
Muscle percentage (%)	42.0	41.8	43.7
Fat percentage (%)	14.8	15.6	12.4
Visceral fat percentage (%)	3	3	3

**Table 2 foods-12-01055-t002:** Dietary sequence by day.

**Test 2. Regular Diet with Ordered Food Consumption**					
**Day**	**Breakfast**	**Snack 1**	**Lunch**	**Dinner**	**Snack 2**
1	D-P-CH	D-CH-FR	VF-CH-P-FT	P-CH-VF-FR	FT
2	P-CH-D	D-CH-FR	CH-P-CH-FT-VF	CH-VF-FR-P	FT
3	D-P-CH	D-CH-FR	P-CH-FT-VF-CH	VF-FR-P-CH	FT
4	P-CH-D	D-CH-FR	CH-FT-VF-CH-P	FR-P-CH-VF	FT
5	D-P-CH	D-CH-FR	FT-VF-CH-P-CH	P-CH-VF-FR	FT
6	P-CH-D	D-CH-FR	VF-CH-P-CH-FT	CH-VF-FR-P	FT
7	D	D-CH-FR	CH-P-CH-FT-VF	VF-FR-P-CH	FT
8	P-CH-D	D-CH-FR	P-CH-FT-VF-CH	FR-P-CH-VF	FT
9	D-P-CH	D-CH-FR	CH-FT-VF-CH-P	FT-CH-VF-FR	FT
10	P-CH-D	D-CH-FR	FT-VF-CH-P-CH	CH-VF-FR-FT	FT
11	D-P-CH	D-CH-FR	VF-CH-P-CH-FT	VF-FR-FT-CH	FT
12	P-CH-D	D-CH-FR	CH-P-CH-FT-VF	FR-FT-CH-VF	FT
13	D-P-CH	D-CH-FR	P-CH-FT-VF-CH	CH-P-VF-FT	FT
14	D	D-CH-FR	CH-FT-VF-CH-P	CH-P-FT	FT
**Test 3. Assisted diet with ordered food consumption**					
**Day**	**Breakfast**	**Snack 1**	**Lunch**	**Dinner**	**Snack 2**
1	CH-P-VF	D-FR	P-VF-CH	CH-P-VF-FT	FR-VF-FT
2	P-CH-VF	D-FR	CH-P-CH-VF	FT-P-CH-VF	FT-FR-VF
3	VF-P-CH	D-FR	CH-VF-P-CH	P-FT-VF-CH	VF-FT-FR
4	VF-P-CH	D-FR	VF-CH-P	VF-CH-FT-P	FR-VF-FT
5	CH-P-VF	D-FR	P-VF-CH	CH-P-D-VF	FR-CH-FT
6	P-CH-VF	D-FR-D	CH-P-VF-CH	VF-D-CH-P	FT-FR-CH
7	VF-P-CH	D-FR-D	VF-CH-P-CH	CH-VF-D-P	CH-FT-FR
8	VF-CH-P	D-FR-D	CH-VF-P	VF-D-P-CH	FR-CH-FT
9	P-D-VF-CH	D-FR	P-CH-VF-CH	CH-D-VF-FT	FR-CH-FT
10	VF-CH-P-D	D-FR	CH-P-CH-VF	VF-FT-CH-D	CH-FR-FT
11	CH-P-D-VF	D-FR	VF-CH-P-CH	FT-D-VF-CH	FT-CH-FR
12	P-D-VF-CH	D-FR-D	P-VF-CH	D-VF-FT-CH	FR-CH-FT
13	VF-P-D-CH	D-FR-D	CH-P-CH-VF	CH-P-VF-FT	CH-FT-FR
14	CH-P-D-VF	D-FR-D	VF-CH-P	CH-D-VF-FT	FT-FR-CH

**Table 3 foods-12-01055-t003:** Proportions of macronutrients ingested.

Aspect Analyzed	Test 1	Test 2	Test 3
Total calories (Cal)	Breakfast: 1079.60 ± 95 Lunch: 919.29 ± 332 Dinner: 707.47 ± 275 Snack 1: 435.66 ± 12 Snack 2: 241.13 ± 158	Breakfast: 332.27 ± 102 Lunch: 1197.30.0 ± 399 Dinner: 572.44 ± 218 Snack 1: 316.50 ± 136 Snack 2: 86.23 ± 30	Breakfast: 335.00 ± 63 Lunch: 490.01 ± 92 Dinner: 387.36 ± 57 Snack 1: 596.21 ± 27 Snack 2: 260.38 ± 44
Carbohydrates (g)	Breakfast: 141.50 ± 14 Lunch: 136.46 ± 85 Dinner: 79.38 ± 36 Snack 1: 36.51 ± 7 Snack 2: 28.03 ± 18	Breakfast: 40.82 ± 7 Lunch: 182.10 ± 70 Dinner: 52.98 ± 13 Snack 1: 40.51 ± 33 Snack 2: 2.69 ± 1	Breakfast: 30.18 ± 8 Lunch: 65.43 ± 7 Dinner: 33.85 ± 5 Snack 1: 73.10 ± 6 Snack 2: 41.50 ± 6
Carbohydrate (Cal) (%) Energy proportion	Breakfast: 566.00 ± 58→ (52 ± 3)% Lunch: 545.86 ± 343 → (55 ± 17)% Dinner: 317.53 ± 146→ (44 ± 7)% Snack 1: 146.04 ± 31 → (33 ± 6)% Snack 2: 112.13 ± 75 → (47 ± 18)%	Breakfast: 162.87 ± 30 → (47 ± 15)% Lunch: 728.40 ± 282 → (58 ± 12)% Dinner: 211.94 ± 54 → (39 ± 11)% Snack 1: 149.74 ± 137 → (40 ± 17)% Snack 2: 10.76 ± 4 (12 ± 3)%	Breakfast: 120.74 ± 33 → (35 ± 5)% Lunch: 261.72 ± 31 → (54 ± 8)% Dinner: 135.40 ± 20 → (34 ± 1)% Snack 1: 292.42 ± 26 → (48 ± 2)% Snack 2: 166.03 ± 26 → (63 ± 1)%
Lipids (g)	Breakfast: 37.30 ± 7 Lunch: 26.96 ± 11 Dinner: 25.22 ± 12 Snack 1: 16.02 ± 0.2 Snack 2: 8.28 ± 5	Breakfast: 7.61 ± 4 Lunch: 33.15 ± 12 Dinner: 26.36 ± 5 Snack 1: 4.25 ± 4 Snack 2: 7.18 ± 2	Breakfast: 14.07 ± 2 Lunch: 8.40 ± 2 Dinner: 15.73 ± 5 Snack 1: 17.32 ± 0.1 Snack 2: 8.59 ± 2
Lipid Calories (Cal) (%) Energy proportion	Breakfast: 335.42 ± 69 → (30 ± 4)% Lunch: 242.70 ± 103 → (30 ± 15)% Dinner: 227.02 ± 109 → (30 ± 6)% Snack 1: 144.26 ± 2 → (33 ± 1)% Snack 2: 74.56 ± 49 → (35 ± 22)%	Breakfast: 68.54 ± 42 → (18 ± 8)% Lunch: 301.14 ± 113 → (27 ± 10)% Dinner: 105.44 ± 23 → (39 ± 17)% Snack 1: 38.26 ± 18 → (12 ± 5)% Snack 2: 64.65 ± 23 → (75 ± 6)%	Breakfast: 126.68 ± 24 → (38 ± 5)% Lunch: 75.64 ± 24 → (15 ± 2)% Dinner: 141.58 ± 45 → (36 ± 7)% Snack 1: 155.95 ± 1 → (26 ± 1)% Snack 2: 77.36 ± 18 → (29 ± 2)%
Protein (g)	Breakfast: 44.53 ± 4 Lunch: 32.68 ± 9 Dinner: 40.72 ± 18 Snack 1: 36.3 ± 3 Snack 2: 13.60 ± 16	Breakfast: 25.26 ± 11 Lunch: 42.11 ± 11 Dinner: 26.36 ± 5 Snack 1: 32.12 ± 6 Snack 2: 2.70 ± 1	Breakfast: 21.89 ± 3 Lunch: 38.16 ± 15 Dinner: 27.59 ± 6 Snack 1: 36.95 ± 0.5 Snack 2: 4.24 ± 1
Protein Calories (Cal) (%) Energy proportion	Breakfast: 178.15 ± 17 → (16 ± 3)% Lunch: 130.73 ± 37 → (14 ± 3)% Dinner: 162.91 ± 75 → (24 ± 9)% Snack 1: 145.3 ± 15.67 → (33 ± 4)% Snack 2: 54.43 ± 67 → (16 ± 15)%	Breakfast: 101.07 ± 46 → (26 ± 10)% Lunch: 167.15 ± 46 → (14 ± 3)% Dinner: 105.44 ± 23 → (21 ± 10)% Snack 1: 128.48 ± 25 → (46 ± 16)% Snack 2: 10.80 ± 6 → (12 ± 4)%	Breakfast: 87.57 ± 14 → (26 ± 3)% Lunch: 152.64 ± 63 → (30 ± 7)% Dinner: 110.3 ± 27 → (28 ± 7)% Snack 1: 147.83 ± 2 → (24 ± 1)% Snack 2: 16.98 ± 4 → (6 ± 1)%

**Table 4 foods-12-01055-t004:** Glucose statistics at the end of each test.

Test Number	Glucose Average (mg/dL)	Glucose Management Indicator (%)	Coefficient of Glucose Variation (%)
Test 1	84.06 ± 13.77	5.32	16.39
Test 2	88.88 ± 14.49	5.43	16.31
Test 3	93.13 ± 11.93	5.53	12.81

**Table 5 foods-12-01055-t005:** Postprandial glucose average at each meal.

Intake	Test 1 (mg/dL)	Test 2 (mg/dL)	Test 3 (mg/dL)
Breakfast	83.02 ± 1.87	87.49 ± 2.28	89.78 ± 2.27
Lunch	90.39 ± 4.02	98.31 ± 5.60	100.37 ± 2.83
Dinner	96.53 ± 4.62	95.10 ± 7.02	100.77 ± 3.58
Snack 1	91.32 ±3.66	89.34 ± 4.59	90.48 ± 2.26
Snack 2	96.96 ± 3.43	87.66 ± 1.81	92.94 ± 3.84

**Table 6 foods-12-01055-t006:** Glycemic variation in relation to the first macronutrient ingested.

First Macronutrient Ingested	Glucose Average (mg/dL)	Maximum Peak (mg/dL)	Peak Time (h)	Stabilization Time (h)
Carbohydrates	Breakfast: 90.72 ± 4 Lunch: 98.48 ± 3 Dinner: 95.40 ± 7	Breakfast: 115.50 ± 11 Lunch: 123.62 ± 12 Dinner: 125.66 ± 20	Breakfast: 3.02 ± 1.2 Lunch: 2.63 ± 1.8 Dinner: 2.36 ± 1.3	Breakfast: 3.54 ± 1.0 Lunch: 3.02 ± 1.2 Dinner: 3.42 ± 0.8
Vegetables and Fiber	Breakfast: 87.81 ± 4 Lunch: 93.67 ± 14 Dinner: 95.74 ± 3	Breakfast: 113.33 ± 5 Lunch: 119.55 ± 16 Dinner: 117.44 ± 6	Breakfast: 3.10 ± 1.0 Lunch: 2.34 ± 1.9 Dinner: 2.80 ± 1.3	Breakfast: 3.99 ± 1.3 Lunch: 2.96 ± 1.1 Dinner: 3.58 ± 0.9
Proteins of animal origin	Breakfast: 87.72 ± 5 Lunch: 99.91 ± 3 Dinner: 82.02 ± 11	Breakfast: 112.24 11 Lunch: 123.14 ± 14 Dinner:107.00 ± 24	Breakfast: 2.82 ± 0.9 Lunch: 2.51 ± 1.5 Dinner: 2.38 ± 1.9	Breakfast: 3.44 ± 1.0 Lunch: 3.70 ± 0.9 Dinner: 3.48 ± 1.0

## Data Availability

The data are available from the corresponding author.

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
