# Peer review of "Impact on Glycemic Variation Caused by a Change in the Dietary Intake Sequence"

_foods, 2023, doi:10.3390/foods12051055_

Round 1
Reviewer 1 Report
The manuscript explained the dietary intake sequence effect on the glycemic index. It is a nutritious work and interesting and relevant for the readers of the nutrition. The topic is novel, however the introduction needs to refine and evaluate more recent work and merge the last two paragraphs. The structure of the M&M section was designed properly, but it requires adding statistical analysis as the main concern of the manuscript and including conflict of interest. Since the contest of the analyst is a necessary part. The results and discussion is proper, although, adding ±SD, such as Table 4 for glucose average is required. Furthermore, glucose average is correct not average glucose!! The conclusions consistent with the evidence and arguments presented and they addressed the main question posed. Finally a grammatical corrections and improvement in the language through the whole manuscript is required.
Reviewer 2 Report
This study showed the effect on glycemic response when macronutrients are consumed in different order by one healthy individual. The results appear to be consistent with other studies though literature on this subject appears to be scant. The complexity of conducting this study is appreciated. However, there needs to be considerably more detail provided in methods so that the process could be replicated. The Discussion needs more organization and, given that there is only one subject, conclusions need to be more restrained. There are a number of incomplete sentences throughout. Some are indicated, but please check the manuscript for others that may have been missed. Comments and suggestions are provided below.
In lines 65-78, it needs to be emphasized that this research includes only one person. Although lines 69-70 do state “for a healthy person”, it reads to me that you are talking more generically about a healthy person.
Lines 76-78 do not represent a complete sentence. Perhaps state “This study will provide representative data of the phenomenon [what phenomenon?] that could potentially be applied to a population, allowing ……”.
The Methods section needs considerably more detail. Conceivably, others might want to replicate this research. Please include more information about the following:
· Was this a feeding study? How was compliance evaluated?
· What instructions was the participant provided?
· What was the percentage of energy intake provided by each macronutrient?
· What was the rationale for having 4 different days of orders of consumption through each test?
· VF indicates Vegetables and Fiber. Was a fiber supplement provided or were certain vegetables, forms, and amounts prescribed to increase fiber intake?
· It would be helpful if there were sample menus or meal plans with food quantities for each test period. Perhaps this could be included as Supplemental tables.
· Foods are often mixtures that contain all three macronutrients and possibly fiber. How were these types of foods handled in the order of introduction?
· Table 3 is titled “Proportions of macronutrients ingested” but only includes Carbohydrate. Please include Protein and Fat. Either instead of or in addition to kcal for each nutrient, percentages from each nutrient would be helpful.
Results
Line 164: What do you mean by “notorious”?
Line 173-175 is not a complete sentence.
Table 5: Are these average pre-prandial glucose readings? Please clarify. The reader should be able to interpret the table without reading the text.
Section 3.2 seems more like Methods, not results.
Line 256-258 is not a complete sentence.
Discussion
The Discussion needs more organization.
Lines 234-240 seem out of place. The Discussion should start with a summary of key findings, then discussion of these findings in light of other studies. For example, those cited in Lines 243-249; references 12 and 27 could be used to compare and contrast results of this study.
Reference 22: Is this a study?
Line 261: I suggest replacing “generates” with “illustrates”.
Line 274: I suggest replacing "demonstrates” with “suggests”.
Line 256-258 and 276-278 are not complete sentences.
Line 250-254: I suggest expanding discussion of this point to include plans and suggestions for future research. The Introduction and Purpose imply that this was a pilot study. What’s next?
Also, include a discussion of strengths and limitations.
Conclusion
This study does not corroborate anything (Line 281). Though there were numerous measurements, it was one person and was consistent with other observations, which you should discuss in the Discussion.
I suggest including a conclusion about the process that was developed (e.g. necessary experimentation – Line 251). Though arduous, do you conclude it’s feasible? Wasn’t this also a primary purpose of the study?
Line 284-285 and 285-287 are not complete sentences.
Reviewer 3 Report
My compliments with the author, glycemic variability il a new concept really important especially in patients with type 2 diabetes. For that reason i believe that introduction must be improved, it will be useful to describe better the defintion of type 2 diabetes, insulin resistance and add the reference "the molecular link between oxidative stress, insulin resistance and type 2 diabetes: a target for new therapies against cardiovascular diseases".
Do you have any data regardign the SGLT2i?
